# Characteristics of TIMP1, CD63, and β1-Integrin and the Functional Impact of Their Interaction in Cancer

**DOI:** 10.3390/ijms22179319

**Published:** 2021-08-27

**Authors:** Beatriz Laís Justo, Miriam Galvonas Jasiulionis

**Affiliations:** Department of Pharmacology, Escola Paulista de Medicina, Universidade Federal de São Paulo (UNIFESP), Rua Pedro de Toledo 669, 5 Floor, São Paulo 04039-032, Brazil; beatrizjusto.bj@gmail.com

**Keywords:** TIMP-1, cancer, *N**-*glycosylation, CD63, β1-integrin

## Abstract

Tissue Inhibitor of Metalloproteases 1, also known as TIMP-1, is named for its well-established function of inhibiting the proteolytic activity of matrix metalloproteases. Given this function, many studies were carried out to verify if TIMP-1 was able to interrupt processes such as tumor cell invasion and metastasis. In contrast, many studies have shown that TIMP-1 expression is increased in several types of tumors, and this increase was correlated with a poor prognosis and lower survival in cancer patients. Later, it was shown that TIMP-1 is also able to modulate cell behavior through the induction of signaling pathways involved in cell growth, proliferation, and survival. The mechanisms involved in the regulation of the pleiotropic functions of TIMP-1 are still poorly understood. Thus, this review aimed to present literature data that show its ability to form a membrane complex with CD63 and β1-integrin, and point to *N**-*glycosylation as a potential regulatory mechanism of the functions exerted by TIMP-1. This article reviewed the characteristics and functions performed individually by TIMP1, CD63, and β1-integrin, the roles of the TIMP-1/CD63/β1-integrin complex, both in a physiological context and in cancer, and the regulatory mechanisms involved in its assembly.

## 1. Introduction

Tissue Inhibitors of Matrix Metalloproteases (TIMPs) are natural inhibitors of Matrix Metalloproteases (MMPs). The balance between the activities of MMPs and TIMPs is essential for extracellular matrix maintenance, remodeling, and turnover in physiological events [1]; however, the imbalance between these two components is associated with a variety of pathological conditions, including cancer [2], since tumor cells need the proteolytic activity of MMPs for processes such as growth, migration, invasion, and metastasis to occur [3]. Thus, many studies have been carried out with TIMPs to prove that these natural inhibitors can modulate the invasive and metastatic capacity of tumor cells [4,5,6]. These promising results have culminated in the development of synthetic metalloprotease inhibitors (MPIs) as a new form of cancer therapy [7]. However, MPIs did not bring the expected results in clinical trials [8]. In addition, other studies have also shown that the expression of TIMPs, especially TIMP-1, is increased in several types of tumors, and this increase in its expression has been correlated with a poor prognosis of these tumors [9,10,11,12,13,14,15,16,17], which goes against the initial concept that TIMPs, including TIMP-1, could prevent tumor progression and metastasis.

The correlation of increased TIMP-1 expression with a poorer prognosis in cancer patients made researchers question the roles that TIMP-1 could play in tumor progression, in addition to its inhibitory function of MMPs. The discovery that TIMP-1 can bind to the cell membrane through CD63 and β1-integrin and activate signal transduction pathways that modulate cell behavior [18]—including proliferation, growth and survival—demonstrated that TIMP-1 plays an important role in the processes involved in tumor progression and metastasis. For this reason, the functions of TIMP-1, both oncogenic (activation of intracellular signaling pathways) and tumor suppressor (MMPs inhibition), as well as the mechanisms involved in the regulation of these functions, should be better understood. Here, we reviewed the characteristics and functions of TIMP-1 and its binding partners, CD63 and β1-integrin, and the evidence indicating that the pleiotropic functions of TIMP-1 could be regulated by the presence of aberrant *N**-*glycosylation patterns.

## 2. Characteristics and Structure of TIMP-1

Tissue Inhibitor of Metalloproteases 1 (TIMP-1) belongs to the family of TIMPs, which members (TIMP-1, 2, 3, and 4) have sequence and structure homology. The TIMP1 gene is located on the human chromosome Xp11.1-p11.4. Its messenger RNA (mRNA) encodes a protein of 184 amino acids, with a molecular mass ranging from 29 to 34 kDa [19].

Like the other members, TIMP-1 has in its composition 12 cysteine residues that form six disulfide bonds that fold the protein into two structural domains: an *N*- and a *C*-terminal domain. Each domain contains 6 cysteine residues forming 3 disulfide loops resulting, at the end, in 6 loops in the TIMP-1 molecule [19] (Figure 1).

As a soluble protein secreted into the extracellular environment, TIMP-1 is found in most tissues and body fluids. TIMP-1 expression can be induced in several cell types, mainly due to external stimuli, such as growth factors (β-FGF, PDGF, EGF), phorbol esters, serum, and cytokines (IL-6 and IL-1) [20]. Furthermore, some studies even show that TIMP-1 expression can be regulated by epigenetic mechanisms, such as DNA methylation and histone deacetylation [16,21].

## 3. TIMP-1 as a Multifunctional Protein

### 3.1. Inhibitory Activity of Matrix Metalloproteases (MMPs)

As their name suggests, TIMPs are natural inhibitors of matrix metalloproteases (MMPs) by inhibiting the proteolytic activity of these enzymes [2]. This inhibition via TIMP-MMP occurs through the interaction of Zn^2+^, present in the MMP catalytic site, with the amino and carbonyl groups of the amino acid residues around the Cys1–Cys70 disulfide bond in the TIMP *N*-terminal region [2]. Therefore, it was shown that the *N*-terminal domain of TIMP-1 is responsible for the inhibitory activity of MMPs.

The first function described for TIMPs was to inhibit MMPs; however, several studies report that TIMPs, and especially TIMP-1, also exhibit functions and activities distinct from the inhibitory activity of MMPs (for more information, see Lambert et al., 2004) [20]. One of the most important roles played by TIMP-1, in addition to inhibiting MMPs, is to induce different intracellular signaling pathways that modulate cell behavior. Later, it is discussed that these multiple activities of TIMP-1 have major implications for physiological and pathological processes.

### 3.2. Activation of Intracellular Signaling Pathways

#### 3.2.1. Cell Growth

TIMP-1 was once known as erythroid-potentiating activity (EPA) for inducing erythroid progenitor cells to grow and proliferate [22]. From these results, other studies aimed to investigate whether TIMP-1 induced cell growth in other cell lines. As expected, TIMP-1 promoted cell growth in a wide variety of normal cells, including human keratinocytes, chondrocytes, fibroblasts, epithelial and endothelial cells, and lymphoid and myeloid cells [23,24].

#### 3.2.2. Cell Survival and Apoptosis

In addition to its cell growth-promoting activity, TIMP-1 can also inhibit apoptosis by activating cell survival pathways. TIMP-1 expression has been associated with resistance to apoptosis in B lymphocyte cell lines, human mammary epithelial cells, and hematopoietic cells [12,13,25].

In a study performed in a series of Burkitt’s lymphoma cell lines, the overexpression of TIMP-1 was shown to suppress apoptosis by inducing the expression of BCL-XL, an important anti-apoptotic protein [12]. In another study in MCF10A and MCF-7 breast epithelial cells, it was demonstrated that the inhibition of apoptosis, mediated by TIMP-1, occurs through the indirect regulation of the activity of anti-apoptotic and pro-apoptotic proteins of the BCL-2 family, since TIMP-1 induces activation of cell survival signaling pathways involving FAK and PI3K-AKT [13]. AKT activation results in BAD phosphorylation, and phosphorylated BAD is no longer able to interact with BCL-2, resulting in the activation of BCL-2 and, consequently, in apoptosis inhibition. In addition, this study also demonstrated that the downregulation of TIMP-1 expression increases cell death by apoptosis [13,26].

Finally, it was shown that TIMP-1 has an anti-apoptotic effect in the human erythroleukaemic cell line UT-7 and in the murine myeloid cell line 32D by inducing survival pathways. TIMP-1 leads to the activation of JAK2 and PI3K signaling pathways, important pathways for AKT activation, responsible for phosphorylating and inhibiting BAD. Phosphorylated BAD leads to increased levels of BCL-XL and, consequently, apoptosis inhibition [25].

#### 3.2.3. Cell Proliferation and Differentiation

In addition to its activities of promoting cell growth and survival and inhibiting apoptosis, TIMP-1 has also been characterized by inducing cell proliferation and differentiation of the UT-7 erythroid cells. TIMP-1 activates these pathways through the P38, MAPK, and JNK activation [27].

This literature proved that TIMP-1 also has the function of activating different signal transduction pathways, independently of its inhibitory activity on MMPs. As the activation of intracellular signaling pathways normally depends on the stimulation of a plasma membrane receptor, it was hypothesized that TIMP-1 could present its own cell membrane receptor.

## 4. Association between TIMP-1 and CD63

As mentioned, several studies have shown that TIMP-1 is a multifunctional protein that has pleiotropic activities independent of MMP inhibition. Among the various functions of TIMP-1, the activation of intracellular signaling pathways involved in cell growth, differentiation, proliferation, survival, and apoptosis is included. These activities were associated with the ability of TIMP-1 to bind to the cell surface, reinforcing that TIMP-1 has its own cell membrane receptor responsible for mediating the activation of different signal transduction pathways.

After many studies, CD63 was finally identified as the membrane receptor for TIMP-1 [18]. The most recent study further demonstrated that TIMP-1 interacts with CD63 through nine amino acid residues present in its *C*-terminal region [28].

## 5. Characteristics and Structure of CD63

The *CD63* gene is located on human chromosome 12q13 and encodes for a 237 amino acid protein with a molecular mass ranging from 30 to 60 kDa [29].

CD63 belongs to the tetraspanins family (also called the 4-transmembrane superfamily—TM4SF), a 28-member superfamily of plasma membrane proteins characterized by their four hydrophobic transmembrane domains. The structure of tetraspanins, including CD63, consists of *N*- and *C*-terminals forming short cytoplasmic domains, as well as an intracellular interconnection loop between transmembrane domains 2 and 3, and two extracellular domains (loops): a small loop between transmembrane domains 1 and 2 (SEL), and a large loop between transmembrane domains 3 and 4 (LEL) [30] (Figure 2).

Tetraspanins are characterized by highly conserved sequences in their four transmembrane domains and by the presence of the CCG motif within the large extracellular loop [30]. However, the *N*- and *C*-terminal of tetraspanins differ between each family member, which may imply different functions for each tetraspanin [31]. In its structure, CD63 is differentiated by carrying the YXXX motif in its *C*-terminal cytoplasmic domain, which contains an essential tyrosine residue (Y) and 3 hydrophobic residues (XXX). The YXXX motif is required for CD63 endocytosis and its targeting to the lysosome [32].

CD63 is expressed on almost all cell and tissues types and is located not only on the cytoplasmic membrane, but also in late endosomes, lysosomes, and multivesicular bodies (MVBs) [33]. In most cells, the CD63 pool is present on the membrane of late endosomes and lysosomes due to the lysosome targeting signal sequence present in its structure (YXXX) [34]. CD63 is also present in MVBs of platelet granules, melanosomes of melanocytes, cytotoxic granules of T cells, Weibel-Palade bodies of endothelial cells, and Major Histocompatibility Complex II (MHC-II) compartments of dendritic cells [35,36,37,38,39]. The stimulation of these cells leads to the fusion of these multivesicular bodies with the cell surface, resulting in the release of microvesicles, called exosomes, in the extracellular microenvironment. For this reason, CD63 is highly enriched in exosomes derived from different cell types [40].

## 6. Functions of Tetraspanins and CD63

The main property of tetraspanins is the ability to interact with each other and with a wide variety of cell surface proteins to form a network of supramolecular complexes [29]. Thus, tetraspanins organize themselves into membrane domains called tetraspanin-enriched microdomains (TEMs) [41,42].

The main role of tetraspanins, when associating with other proteins forming supramolecular complexes on the cell surface, is to trigger signal transduction pathways [43]. These supramolecular complexes influence cellular behavior, mainly in basic cellular processes, such as cell proliferation, migration, adhesion, differentiation, and motility [31,44]. The functions of tetraspanins, that is, the activation of specific signaling pathways, depend on the associated proteins and cell types involved.

CD63 was the first characterized tetraspanin [45], and was originally known as platelet glycoprotein 40 (Pltgp40) due to its location on the cell surface of activated platelets collected from human blood [46]. It was later shown that CD63 is transferred to the cell membrane after platelet activation, where it associates with and modulates the platelet integrin αIIBβ3-CD9 complex [47].

As already mentioned, CD63 has also been described as located in the Weibel-Palade bodies (WPB) of endothelial cells [38]. Recently, it was shown that CD63 is an essential cofactor for the trafficking of P-selectin from WPB to the plasma membrane, since mice deficient in CD63 showed absence of P-selectin on the surface of endothelial cells, demonstrating that CD63 plays a crucial but indirect role in the recruitment, scrolling, and extravasation of leukocytes during the inflammatory process [48].

CD63 was also found to be associated with β1-integrin and VEGFR2 in the plasma membrane of human umbilical vein endothelial cells (HUVECs), where it acts by modulating intracellular signaling induced by β1-integrin, since the loss of CD63 expression led to a decrease in FAK phosphorylation. The CD63/β1-integrin/FAK pathway is an important signaling pathway involved in the migration and adhesion of these cells during the angiogenesis process [49].

Finally, it was observed that CD63 is also transferred to the plasma membrane of T lymphocytes after its stimulation, where it acts as a co-stimulatory protein, since the co-stimulation of CD63 and TCR induced the activation and proliferation of T cells, as well as IL-2 production [50].

## 7. Binding Partners of Tetraspanins and CD63

Tetraspanins can form supramolecular complexes with several cell surface proteins, such as integrins, membrane receptors, intracellular signaling molecules, and even with other tetraspanins; however, the best-elucidated binding partner of tetraspanins are integrins, since these integrin-tetraspanin complexes are commonly detected in different cell types [51,52,53,54,55]. Furthermore, the most-characterized integrin-tetraspanin interactions are those involving β1-integrin, since interactions between β1-integrin and different members of tetraspanins in various cell types have already been described [56]. Numerous studies have shown that these integrin-tetraspanin complexes are extremely important for the regulation of cell migration and adhesion [57,58,59,60].

As a tetraspanin, CD63 can also interact with a wide variety of proteins, directly or indirectly, as integrins (mostly β1-integrin), other tetraspanins, cell surface receptors, adapter proteins, intracellular signaling molecules, and lysosomal proteins, among other proteins [61,62,63,64,65].

As mentioned earlier, CD63 has also been identified as a TIMP-1 receptor, together with β1-integrin, to form a supramolecular complex located on the surface of human breast epithelial cells [18]. The most recent study further demonstrated that CD63 only interacts with TIMP-1 through its large extracellular loop [28].

## 8. Characteristics and Structure of Integrins

Integrins are transmembrane proteins formed by two non-covalently associated heterodimeric subunits: one of the 8 β subunits combines with one of the 18 α subunits [66]. There are at least 24 integrins already described. Of these, half contain the β1 subunit [66]. β1-integrin is the most abundant subunit, as expressed by a variety of cell types and present in almost all tissues [67]. The *ITGB1* gene is located on human chromosome 10p11.22 and its messenger RNA (mRNA) encodes a protein of approximately 798 amino acids and molecular mass ranging from 100 to 132 kDa [68].

The two subunits have a small cytosolic *C*-terminal tail and a large extracellular *N*-terminal domain [69] (Figure 3). The *N*-terminal extracellular domains of integrins are responsible for binding to extracellular matrix components or, in some cases, to cell surface ligands [69]. The *C*-terminal intracellular domains of integrins bind to cytoplasmic adapter proteins to anchor cytoskeletal actin filaments [70], and to cytoplasmic kinases to activate intracellular signaling cascades [71].

## 9. Functions of Integrins

### 9.1. Cytoskeleton Rearrangement and Cell Motility

Integrins are the main membrane receptors responsible for cell adhesion to extracellular matrix components (cell-matrix contact); however, integrins have also been observed interacting with cell surface ligands (cell-cell contact) and with soluble ligands [66].

In order to make contact between the cell membrane and extracellular matrix, cells form specialized structures called focal adhesions. Focal adhesions are formed by integrins, adapter molecules, intracellular signaling proteins, and cytoskeleton filaments. When integrins interact with extracellular matrix components, recruitment of focal adhesion components occurs. Talin and FAK are the first proteins to be recruited after integrin activation. FAK phosphorylation/activation leads to the recruitment of cytoskeletal components, such as vinculin, talin, α-actinin, and actin filaments, in addition to SRC and Rho-like GTPases, which allow the rearrangement of the actin cytoskeleton [72,73]. The formation of focal adhesions with subsequent rearrangement of the actin cytoskeleton confers motility to the cells.

### 9.2. Integrin-Induced Signal Transduction Pathways

In addition to inducing cytoskeleton rearrangement required for cell motility, focal adhesions are also important for activating signal transduction pathways. The best-studied signaling pathway induced by integrins also requires FAK recruitment and phosphorylation. FAK is the main phosphorylated kinase in focal adhesions, since its highly phosphorylated structure allows the formation of binding sites for several intracellular signaling proteins, such as SHC, PLC, and GRB2, responsible for activating MAPK, JNK, and PI3K pathways, which are important signaling pathways involved in cell proliferation and survival [74,75]. Certain integrins can also activate the BCL-2 pathway to prevent apoptosis [76].

Thus, integrins also play a crucial role in activating signal transduction pathways responsible for cellular processes, such as proliferation, growth, migration, survival, and apoptosis [77]. This can be explained by the fact that certain cell types, such as cells of epithelial origin, depend on adhesion to the extracellular matrix to grow, proliferate, and survive. This dependence is known as anchorage dependence, being mediated by intracellular signals generated by integrins. Anchorage dependence can be seen in cells that undergo apoptosis when they lose contact with the extracellular matrix—cell death known as *anoikis* [78]. Therefore, intracellular signals produced in focal adhesions and mediated by integrins are crucial for growth, proliferation, survival, and resistance to *anoikis* [77,78].

## 10. Integrin Binding Partners

### 10.1. Extracellular Binding Partners

The main extracellular ligands that interact with integrins are the components of the extracellular matrix, which include collagen, laminin, fibronectin, and vitronectin [68], since the primary function of integrins is to anchor cells to the matrix extracellular; however, integrins also carry out cell-cell interactions. For example, during the rolling and infiltration of neutrophils into inflammation sites, neutrophils need to adhere to vascular endothelial cells, and this adhesion is mediated by the binding between integrins present on the neutrophil cell surface, and ICAM-1, present on the endothelial cell surface [79].

### 10.2. Cytoplasmic Binding Partners

Integrins need different binding partners to perform their functions, such as proteins that anchor integrins to the cytoskeleton and serve as anchoring sites for signaling molecules such as α-actinin, talin, and vinculin [73]. In addition, the cytoplasmic domains of integrins do not have intrinsic kinase activity; therefore, cytoplasmic kinases such as ILK and FAK are recruited to carry out the phosphorylation function and activate signaling proteins and thus trigger signal transduction pathways [71,80].

### 10.3. Cell Membrane Binding Partners

In addition to interacting with extracellular components and cytosolic proteins, integrins can also horizontally associate with other plasma membrane proteins. Among them, the proteins of the tetraspanin family stand out, since these integrin-tetraspanin complexes have been identified in different cell types [31,81].

As mentioned earlier, the main role of tetraspanins is to associate with other membrane proteins and modulate the intracellular signals generated by them [43]. Thus, it was hypothesized that integrin-induced signaling can be modulated by tetraspanins, since they participate in cell motility [60,82,83]. In addition, other studies have also shown that tetraspanin-associated integrins can recruit cytoplasmic kinases and form signaling complexes [81].

Several integrin-tetraspanin complexes have already been described [84]. Among them, CD63 tetraspanin has been characterized as a binding partner of several integrins, but mainly of β1-integrin [85]. Furthermore, this complex involving β1-integrin and CD63 has been identified on the surface of different cell types [51].

## 11. TIMP-1/CD63/β1-Integrin Complex Formation and Signal Transduction Pathways Involved

The fact that CD63 is characterized as a β1-integrin binding partner, together with the discovery of its role as a membrane receptor for TIMP-1, raised suspicions that CD63, β1-integrin, and TIMP-1 could form a membrane complex. The first time that the TIMP-1/CD63/β1-integrin complex was described was in 2006, when Jung and collaborators identified CD63 as a binding partner of TIMP-1. In this study, researchers observed that TIMP-1 and CD63, together with the β1-integrin molecule, form a supramolecular complex located on the surface of the human breast epithelial cells MCF10A [18]. The TIMP1/CD63/β1-integrin complex was related to the activation of cell survival signaling pathways, mediated by FAK and ERK, and to the inhibition of the pro-apoptotic signaling pathway, involving caspase-3 [18]. From these results, other studies were carried out to better understand its function and implications.

Another study also showed that β1-integrin and CD63 form a complex with TIMP-1 on the cell surface of human CD34⁺ hematopoietic stem and progenitor cells (HSPCs). Through this complex, TIMP-1 activates CD63/β1-integrin in order to increase adhesion and migration of HSPCs, as well as protect them from induced apoptosis, through the MAPK pathway and the Wnt/β-catenin signaling pathway [86].

The TIMP-1/CD63/β1-integrin complex has also been reported in the membrane of oligodendrocyte progenitor cells. In the study, TIMP-1/CD63/β1-integrin signal was used to activate AKT and promote β-catenin signaling, contributing to oligodendrocyte differentiation and maturation and promoting CNS myelination [87].

Recently, it was shown that the assembly of the TIMP-1/CD63/β1-integrin complex on the membrane of dendritic cells challenged with Toxoplasma gondii. In this study, the TIMP1/CD63/β1-integrin complex induced the FAK-mediated signaling pathway, triggering hypermotility and hypermigration [88].

A recent study demonstrated that, when treated with recombinant TIMP-1 (rTIMP1), human brain microvessel endothelial cells exposed to hypoxia and inflammation damage retain the integrity and rigidity of the junctional proteins. This occurs due to the interaction of TIMP-1 with CD63/β1-integrin complex and subsequent activation of FAK-mediated signaling, resulting in decreased RhoA activation and F-actin depolymerization for structural stabilization of endothelial cells. Thus, this study identified a new role for the TIMP-1/CD63/β1-integrin complex in regulating the integrity of the endothelial barrier and, consequently, in protecting the blood-brain barrier [89].

## 12. Discovery of TIMP-1/CD63/β1-Integrin Complex in Cancer and Its Role in Tumor Progression

Early cancer studies aimed to focus on the role of matrix metalloproteases (MMPs), as the proteolytic activity of these enzymes mainly contributes to tumor cell metastasis [2]. In this sense, it was expected that both the expression and the activity of MMPs prevailed compared with TIMP1. However, interestingly, many studies have shown that TIMP-1 expression is also increased in several tumor types, and increased TIMP-1 expression was correlated with poor prognosis of different types of human cancers [9,10,11,12,13,14,15,16,17].

Initially, these data seemed unexpected, but today it is known that TIMP-1 is a multifunctional protein that can promote proliferation, growth, survival, regulate differentiation, and inhibit apoptosis in several tumor types, including Non-Small Cell Lung Cancer, osteosarcoma (MG-63 cell line), colorectal, human gastric cancer, and breast cancer (T-47D, MCF-7, and MDA-MB-231 cell lines) [10,11,14,15,17,90,91,92,93].

As TIMP-1 is only able to trigger intracellular signaling pathways and modulate cell behavior through its interaction with CD63 and β1-integrin, the formation of TIMP-1/CD63/β1-Integrin complex became an interesting target for cancer research. As expected, this complex was observed in different tumor types [94,95,96,97].

One study has shown that brain tumor tissue, which makes up intracranial glioma, secretes TIMP-1 to attract human neural stem cells (hNSCs). In this study, the researchers demonstrated that TIMP-1 binds to CD63 present on hNSCs cell surface to activate β1-integrin-mediated signaling, leading to AKT and FAK activation and a subsequent change in vinculin and F-actin distribution. Consequently, there was an increase in cytoskeleton reorganization and in the number of focal adhesions, resulting in a significant increase in the migration of hNSCs towards the intracranial glioma [94]. It is believed that the tropism of NSCs for lesions in the central nervous system, including brain tumors such as intracranial glioma, occur as an attempt to repair tissue damage caused by the tumor. This study demonstrated that TIMP-1, secreted by the glioma, is an important chemoattractant of NSCs and that the TIMP-1/CD63/β1-integrin complex has an important role in inducing NSC migration. According to the authors, studying the expression of factors involved in the tropism of hNSCs during a pathological process may allow the development of new tumor therapeutic strategies for tissue repair.

Another study also showed that the TIMP-1/CD63/β1-integrin supramolecular complex activates YAP/TAZ transcription factors through the activation of SRC, RhoA, and F-actin assembly with subsequent inactivation of LATS1/2. Complex-mediated activation of YAP/TAZ resulted in cell proliferation of different tumor cell lines, including oral squamous cell carcinoma (HSC2 cell line), cervical squamous cell carcinoma (HeLa cell line), breast cancer (MCF-7 cell line), and osteosarcoma (U2-OS cell line) [95]. The discovery that the TIMP-1/CD63/β1-integrin complex activates the YAP/TAZ signaling pathway to promote the proliferation of different types of cancer cells gives us new insights to understand the molecular pathways involved in the oncogenic role of this complex, in addition to demonstrating that the TIMP-1/YAP/TAZ axis might be considered a new therapeutic target for cancer patients.

Previous data from our laboratory have also shown that TIMP-1 associates with CD63 and β1-integrin to form a supramolecular complex on melanoma cell surface (4C11- and 4C11+ melanoma cell lines) [96]. It was also observed that this TIMP-1/CD63/β1-integrin complex contributes to the acquisition of *anoikis* resistance and cell survival through the activation of the PI3K signaling pathway [97]. The most recent study demonstrated that Vemurafenib-resistant melanoma cells (Mel28 and A375 cell lines) also form the TIMP1/CD63/β1-integrin supramolecular complex. In this study, it was confirmed that these melanoma cells use the TIMP1/CD63/β1-integrin complex to induce the activation of signaling pathways involving NF-κB and ERK precisely to promote resistance to Vemurafenib [98]. Because melanoma is a tumor extremely resistant to treatments, these studies provide crucial information to understand the molecular mechanisms involved in the melanoma cell survival and resistance, and to develop new, truly effective therapeutic strategies against melanoma.

All these studies corroborate the idea that the TIMP-1/CD63/β1-integrin complex modifies the behavior of different types of tumor cells, giving them the ability to migrate, proliferate, and survive (Figure 4). Thus, these results reinforce the importance of studying this complex, and all the molecular mechanisms involved, so that in the future new therapeutic strategies might be developed.

## 13. *N*-Glycosylation as a Possible Regulator of TIMP-1/CD63/β1-Integrin Complex Formation in Cancer

The vast majority of proteins produced by cells undergo post-translational modifications in their structure, with *N*-glycosylation being the most common modification [99]. *N*-Glycosylation is characterized by the addition of oligosaccharide branches to the amino groups present in the side chain of asparagine residues of a protein [100]. *N*-Glycosylation is a very important post-translational modification to establish the structure and function of several proteins and, for this reason, has roles in many physiological processes [101].

At the protein level, *N*-glycosylation allows for proper folding, increased stability and solubility, transport, and correct protein activity; at the molecular level, it enables interactions between proteins and formation of molecular complexes. Yet, at the cellular level, *N*-glycosylation allows cell-cell recognition and cell-matrix interactions [102].

Given the importance of protein *N*-glycosylation, it is not surprising that a small modification in the structure of *N*-glycans would be enough to alter the function of a glycoprotein, or even modify the behavior of the entire cell. For this reason, many *N*-glycosylation alterations have already been reported and related to several pathological processes, mainly with cancer [103,104,105,106,107].

These altered glycosidic branches have already been observed on the cell surface of different tumor types, including human breast, prostate, colorectal, brain, and pancreatic ductal cancer [108,109,110,111,112]. Altered *N*-glycans present on the cell surface (known as aberrant *N*-glycosylation patterns) influence cell behavior with respect to cell growth, proliferation, survival, and invasion [113,114,115] and, for this reason, are associated with the capacity for neoplastic transformation, progression, and metastasis [116].

Several glycosyltransferases are involved in the addition of these aberrant *N*-glycosylation patterns. However, many studies have demonstrated the involvement of β1-6-*N*-acetylglucosaminyltransferase V (GnT-V) in malignant transformation and tumor progression in different cancer types [117,118,119,120]. GnT-V, present in the trans-region of the Golgi complex, acts by adding an *N*-acetylglucosamine (GlcNAc) to a mannose at a β1-6 bond [121]. The presence of this β1,6-GlcNAc branch has been correlated with increased metastatic potential of tumor cells [113,122,123,124].

Numerous glycoproteins can undergo aberrant *N*-glycosylation [116]. CD63, β1-integrin, and TIMP-1 are among glycoproteins already identified as targets for aberrant *N*-glycosylation in cancer [125,126,127].

CD63 has three *N*-glycosylation sites present in the large extracellular-loop domain [34] (Figure 2). One study correlated CD63 *N*-glycosylation, mediated by the enzyme RPN2, with the chemoresistance and invasiveness of MCF-7 breast cancer cells [126]. However, the knowledge about the role of CD63 *N*-glycosylation in cancer is still limited.

The β1-integrin molecule, on the other hand, has 12 *N*-glycosylation sites [128,129], and this modification is fundamental for its correct structural conformation, for its proper transport to the cell surface and, consequently, to perform their functions properly, especially with regard to cell adhesion and migration [128,130]. Due to the large number of *N*-glycosylation sites, β1-integrin is among the main cell surface molecules that carry aberrant *N*-glycosylation patterns, mainly the β1,6-GlcNAc branch caused by the enzymatic activity of GnT-V [127,131,132]. The presence of this aberrant *N*-glycosylation on β1-integrin has been associated with increased migratory and invasive capacity of different tumor cell lines [133,134].

Finally, TIMP-1 can also be glycosylated in the two *N*-glycosylation sites present in its *N*-terminal domain (Asp30 and Asp78) (Figure 1). Glycosylation allows the correct conformation of its protein structure, the increase of its stability, and its transport to the extracellular environment, in addition to exerting an effect on its activities; however, glycosylation of all sites is not necessary for TIMP-1 to perform its functions [19]. TIMP-1 has been identified as a target of aberrant *N*-glycosylation by the action of *N*-acetylglucosaminyltransferase V in human WiDr colon cancer cells, and this aberrant TIMP-1 glycoform has been associated with tumor cell invasion in vitro and metastasis in vivo [125]. The role of aberrant TIMP-1 *N*-glycosylation was later confirmed with the study that showed that these colon cancer cells transfected with an aglycosylated TIMP-1 mutant show a lower rate of cell proliferation, less resistance to apoptosis, and retarded tumor growth [135].

This last result supports the idea that aberrant *N*-glycosylation, specifically, the presence of the terminal branch β-1,6-*N*-acetylglucosamine (β-1,6-GlcNAc), added by the enzyme *N*-acetylglucosaminyltransferase-V (GnT-V), is essential for TIMP-1 to be able to activate signal transduction pathways that induce cell proliferation, growth, and survival and culminate in tumor progression. As mentioned earlier, the TIMP-1 activity is strongly related to its ability to bind to its CD63/β1-integrin receptors. Therefore, adding this information, and knowing that *N*-glycosylation allows the interaction between proteins and the formation of molecular complexes, we raise the hypothesis that the presence of aberrant *N*-glycosylation modulates the function of TIMP-1, either by the complex formation TIMP-1/CD63/β1-integrin or by the inhibitory activity of MMPs. 

This hypothesis is supported by the fact that TIMP-1 associates with CD63 through its *C*-terminal domain and exerts its inhibitory role on MMPs through its *N*-terminal domain. TIMP-1 *N*-glycosylation sites are located in the *N*-terminal domain; thus, the presence of aberrant N-glycosylation could cause an allosteric impediment that would make the association of TIMP-1 with MMPs impossible, and at the same time as the *C*-terminal domain of TIMP-1 would be exposed, allowing the association of TIMP-1 with CD63. 

However, there are still no studies that demonstrate, in detail, how these glycosidic branches impact the conformation of TIMP-1 and its activity to form the supramolecular complex with CD63 and β1-integrin to induce signal transduction pathways. Thus, more studies are needed to confirm this hypothesis.

## 14. Discussion and Future Perspectives

Although some studies show the effect of aberrant *N*-glycosylation, specially β1,6-GlcNAc branches, in TIMP-1, CD63, and β1-integrin, its effect on the formation and the function of the TIMP-1/CD63/β1-integrin complex is still unknown. Inserting mutations in TIMP-1, CD63, and β1-integrin to prevent them from *N*-glycosylation would allow us to understand the effects of *N*-glycosylation on these proteins and on the complex formation and activation of transduction pathways involved in the tumor development and progression.

Investigating the molecular mechanisms involved in the formation of the TIMP-1/CD63/β1-integrin complex would bring insights into the molecular biology of cancer. Studying, in detail, the composition of the glycosidic branches present in these three proteins could provide not only new potential biomarkers but, mainly, also enable the development of new therapies against cancer. One strategy could be to prevent the *N-*glycosylation of these proteins, as well as the formation of this complex and activation of intracellular signaling pathways involved in tumor progression. Another strategy could be targeting a drug only to cells that have aberrant *N*-glycosylation on their surface, without harming healthy cells. In summary, identifying which *N*-glycans TIMP1, CD63, and β1-integrin bear in normal and tumor cells and understanding their role in the complex formation and aggressive tumor traits might provide an opportunity to develop novel therapeutic strategies, and might also be useful as potential biomarkers for tumor diagnosis and/or prognosis. 

## Figures and Tables

**Figure 1 ijms-22-09319-f001:**
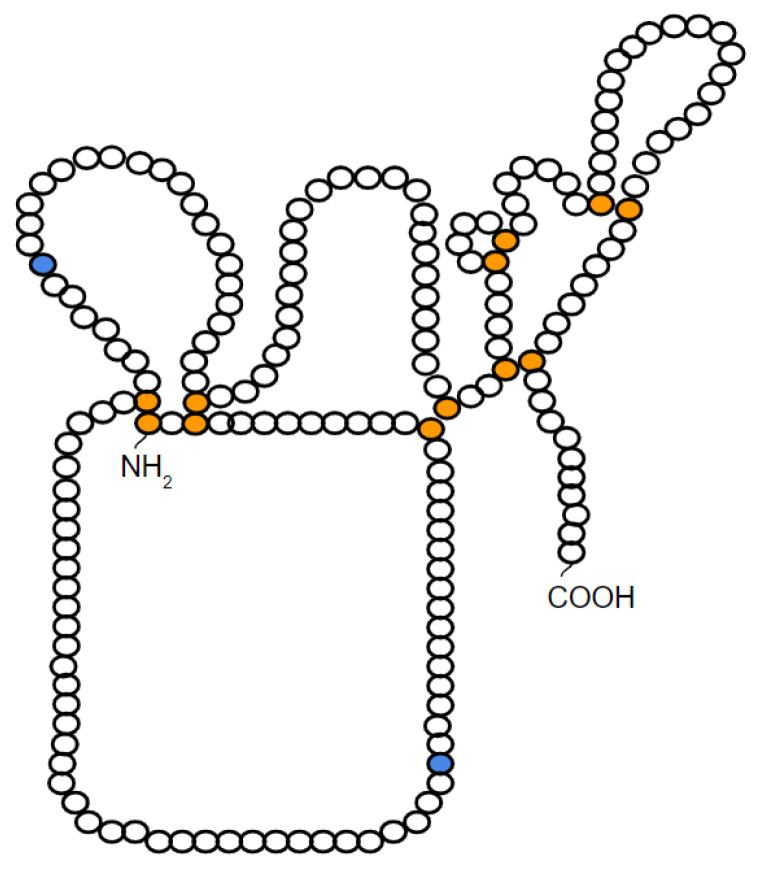
Schematic representation of the probable structure of TIMP-1. In orange, the cysteine residues that form the 6 disulfide bridges are depicted. In blue, the *N*-glycosylation sites present in TIMP-1 are represented. Figure modified from Caterina et al. [19].

**Figure 2 ijms-22-09319-f002:**
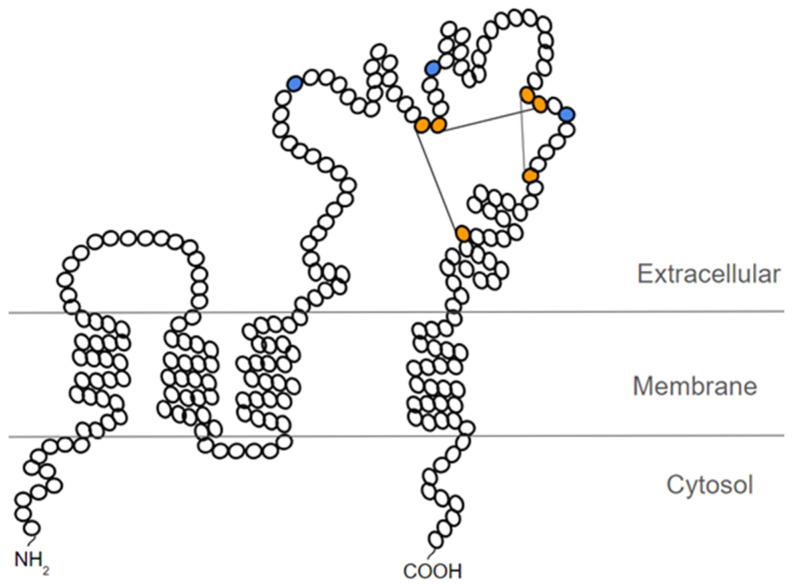
Schematic representation of the probable structure of CD63. In orange, the cysteine residues that form the 3 disulfide bridges are depicted. In blue, the *N*-glycosylation sites present in the large extracellular loop of CD63 are represented. Figure based on the representation proposed by Warner et al. [28].

**Figure 3 ijms-22-09319-f003:**
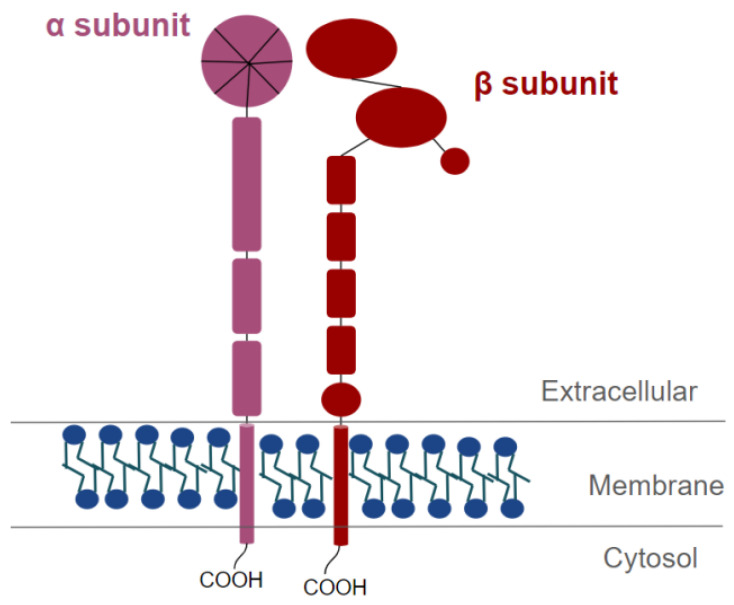
Schematic representation of an integrin heterodimer. In purple, the α subunit is represented, and in red the β subunit is represented.

**Figure 4 ijms-22-09319-f004:**
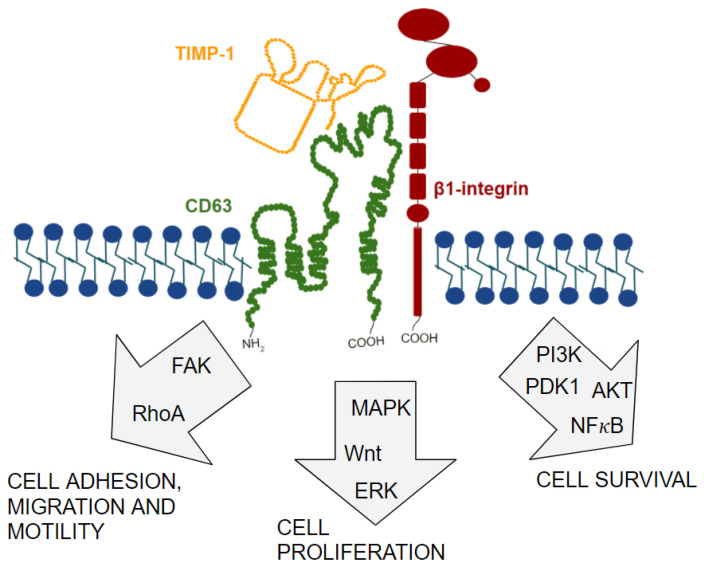
Different intracellular signaling pathways activated by the TIMP-1/CD63/β1-integrin supramolecular complex. Activation of these pathways modulates cell behavior, including cell adhesion, migration, proliferation, and survival.

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
