# Peer review of "Characteristics of TIMP1, CD63, and β1-Integrin and the Functional Impact of Their Interaction in Cancer"

_ijms, 2021, doi:10.3390/ijms22179319_

Round 1
Reviewer 1 Report
The review paper by Justo and Jasiulionis, entitled "N-Glycosylation as a potential regulatory mechanism of TIMP1 function in cancer" gathers the research data on tissue inhibitor of metalloproteinases 1 (TIMP-1) structure and its role, not only as an inhibitor of matrix metalloproteinases (MMPs), but also others less known activities of TIMP-1 which all conform that TIMP-1 is a multifunctional protein. The two groups of TIMP-1 molecular partners, tetraspanins (especially CD63) and integrins (especially β1-integrin) are also characterized. The manuscript is quite interesting but in my opinion its content does not exactly fit to its title. The role of TIMP-1 N-glycosylation in cancer exposed in the title of the review, is only a speculation/hypothesis described shortly at the end of the manuscript which needs verification. I suggest to broaden the title to cover all topics described in the manuscript. Moreover, Authors wrote "N-Glycosylation is essential for TIMP-1 to be able to activate signal transduction pathways that induce cell proliferation, growth and survival and culminate in tumor progression" (lines 437-439), but they did not describe the details of the importance of N-glycosylation in TIMP-1 activity. Which structures are attached to both N-glycosylation sites on TIMP-1 and how these structures impact on signal transduction.
The minor suggestions I listed below:
(1) The names of cell lines should be included when Authors refer to in vitro studies.
(2) Because it is clear that the subsections on page 3 are related to TIMP-1, I suggest to remove TIMP-1 from the title of subsections.
(3) MGAT5/Mgat5 is a gene which encodes β1-6-N-acetylglucosaminyltransferase V (GnT-V). MGAT5 is not the same as GnT-V (lines: 404-405).
(4) It is not clear what decided about giving the full names or using abbreviations, namely:
- some abbreviations are explained only in the text, e.g. EPA (erythroid potentiating activity),
- the other - only in the section Abbreviation at the end of the manuscript, e.g VEGFR, ICAM-1,
- the full names of some of them are given both in the text and the section Abbreviations, e.g. MVBs (lines: 158-159, 161, 485), TEMs (lines: 171-172, 492),
- there are also abbreviations not explained at all (e.g. BCL, BAD).
In the section Abbreviations it is enough to give the general abbreviation of interleukin (IL) instead of repeating of the full names of three interleukins with different numbers.
(5) In the citations added to the legends to Figures 1 and 2 the co-workers should be included: Warner et al. (line 149), Seow et al. (line 235).
(6) In the text "N" from nitrogen in the amino group of asparagine in the name "N-glycosylation" should be italicized as it was done in the title of the manuscript.
(7) The names of genes should be italicized, e.g TIMP1 gene (line 54), CD63 (line 136), β1-integrin gene (line 22).
(8) Other small mistakes should be corrected, e.g.:
- Cys1-Cys70 instead of cys1-cys-70 (line 76),
- Zn2+ instead of Zn2+ (line 75).
Author Response
Please, see the attachment.

Reviewer 2 Report
The manuscript provides a comprehensive review of the potential pathological role of TIPM-1/CD63/ß-1 integrin in cancer progression. Overall, the manuscript is well-written and some minor revisions are suggested.
- It is suggested that the authors should clearly describe the impacts of cited references. For example, what is the consequence or impact of the attraction of neural stem cells by glioma cells through the TIMP-1/CD63/β1-Integrin complex? A similar issue also applies to the activation of YAP/TAZ and the inactivation of LATS1/2 by this complex.
- The section of “Discussion and future perspective” is weak. The authors should provide the experimental suggestions for future investigations in the role of N-glycosylation for oncogenic functions of TIMP-1/CD63/ß-1 integrin complex and the suggestions for the development of therapeutic strategies based on these findings.
Author Response
Please, see the attachment.

Round 2
Reviewer 1 Report
All my suggestions were included in the revised manuscript by Justo and Jasiulionis. It is ready for publication.